# Triptonide is a reversible non-hormonal male contraceptive agent in mice and non-human primates

Zongliang Chang[1,7], Weibing Qin[2,3,7], Huili Zheng[1,7], Kathleen Schegg[1,7], Lu Han[2,3], Xiaohua Liu[2,3], Yue Wang[1], Zhuqing Wang [1], Hayden McSwiggin[1], Hongying Peng[1], Shuiqiao Yuan[1], Jiabao Wu[2,3], Yongxia Wang[2,3], Shenghui Zhu[2,3], Yanjia Jiang[2,3], Hua Nie[2,3], Yuan Tang[2,3], Yu Zhou[2,3], Michael J. M. Hitchcock[4], Yunge Tang[2,3✉] & Wei Yan [1,5,6✉]

There are no non-hormonal male contraceptives currently on the market despite decades of efforts toward the development of "male pills". Here, we report that triptonide, a natural compound purified from the Chinese herb *Tripterygium Wilfordii Hook F* displays reversible male contraceptive effects in both mice and monkeys. Single daily oral doses of triptonide induces deformed sperm with minimal or no forward motility (close to 100% penetrance) and consequently male infertility in 3–4 and 5–6 weeks in mice and cynomolgus monkeys, respectively. Male fertility is regained in ~4–6 weeks after cessation of triptonide intake in both species. Either short- or long-term triptonide treatment causes no discernable systematic toxic side effects based on histological examination of vital organs in mice and hematological and serum biochemical analyses in monkeys. Triptonide appears to target junction plakoglobin and disrupts its interactions with SPEM1 during spermiogenesis. Our data further prove that targeting late spermiogenesis represents an effective strategy for developing non-hormonal male contraceptives.

[1] Department of Physiology and Cell Biology, University of Nevada, Reno School of Medicine, Reno, NV, USA. [2] NHC Key Laboratory of Male Reproduction and Genetics, Guangzhou, People's Republic of China. [3] Family Planning Research Institute of Guangdong Province, Guangzhou, People's Republic of China. [4] Department of Microbiology and Immunology, University of Nevada, Reno School of Medicine, Reno, NV, USA. [5] The Lundquist Institute for Biomedical Innovation at Harbor-UCLA Medical Center, Torrance, CA, USA. [6] Department of Medicine, David Geffen School of Medicine at UCLA, Los Angeles, CA, USA . [8] These authors contributed equally: Zongliang Chang, Weibing Qin, Huili Zheng, Kathleen Schegg. ✉email: tyg813@126.com; wei.yan@lundquist.org

Overpopulation and unintended pregnancy underscore a critical need for next-generation contraceptives that should be safe, convenient, effective, affordable, and acceptable to people of various cultural and religious backgrounds. Among all of the currently available contraceptives, "oral pills" remain the most popular method[1]. However, contraceptive pills are now only available for women. Despite five decades of efforts, there remain no nonhormonal male birth control pills on the horizon. Failure to develop nonhormonal male contraceptives stems, in part, from our incomplete understanding of spermatogenesis and sperm biology. For example, many believe that male pills should suppress sperm counts to very low or even zero to prevent pregnancy. However, a total blockage of sperm production requires the depletion of spermatogenic cells, which often causes testis shrinkage, an undesirable effect that may deter its usage. Moreover, germ cell depletion alters the cellular composition and microenvironment in the testis, which, in turn, tend to trigger the hypothalamus–pituitary–testis feedback system, leading to systematic side effects[2,3]. Therefore, it remains challenging to identify a compound which can eliminate all sperm without causing toxic side effects. We proposed a strategy for developing male contraceptives, i.e., disabling, instead of depleting, spermatogenic cells or sperm by causing sperm deformation and/or dysfunction[4]. This idea was inspired by several decades of studies on genes encoding proteins exclusively expressed in elongating and elongated spermatids, e.g., *Prm1, Tnp1, Spem1, Catsper1-4, Meig1*, etc., using gene knockout (KO) mouse models[4]. Although many of these KO males are completely infertile, their testis weight, sperm counts, and even testicular histology are largely normal, and infertility of these mice results from either sperm deformation (e.g., teratozoospermia in *Spem1-* or *Meig1-null* mice)[5], or lack of functional components (e.g., absence of an ion channel in *Catsper3-* or *Catsper4-*null sperm)[6,7]. Given that proteins encoded by these genes are exclusively expressed in late spermatids, these KO studies strongly suggest that late spermiogenesis appears to lack a stringent "checkpoint" for eliminating defective late spermatids[4]. Consequently, there are typically no histologically discernable disruptions in the seminiferous epithelium although defective sperm are made in these KO mice. The lack of such a "quality control" mechanism in late spermiogenesis is also supported by the fact that a significant proportion of spermatozoa collected from the mouse epididymis (~20–30%) or human ejaculates (30–40%) are immotile and/or morphologically abnormal[8,9]. Based on these observations, we proposed that gene products (e.g., proteins and RNAs) that are exclusively expressed in late spermatids and play an essential role in normal sperm production and male fertility represent ideal male contraceptive targets because a drug that acts on these targets would cause deformed and/or nonfunctional sperm that are incompetent for fertilization without causing significant decrease in either testis size or sperm counts[4].

With this idea in mind, we embarked on an extensive search for known drug candidates that have been documented to cause sperm deformation as a side effect. During this process, we identified triptonide, a compound purified from the extracts of a Chinese herb called *T. wilfordii Hook F* (Supplementary Fig. 1), as a promising nonhormonal male contraceptive agent. This herb has been used for more than two centuries in traditional Chinese medicine to treat a variety of autoimmune and inflammatory diseases, including rheumatoid arthritis[10,11]. However, it was first reported in 1983 that men taking this herbal mixture as medicine for an extended period (>3 months) displayed infertility due to deformed sperm and reduced sperm counts and motility[12]. Since then, researchers have been isolating and testing compounds purified from this herb, in the hope of identifying compounds that have "antisperm effects"[12–18]. To date, hundreds of individual compounds have been purified from this herb, including triptolide, tripdiolide, triptolidenol, tripchlorolide, 16-hydroxytriptolide, triptonide, and many more[19]. In particular, several initial studies on two of these compounds—triptolide and tripchlorolide—reported severe liver toxicity and limited reversibility of male fertility at doses that can effectively reduce sperm count and motility[20–24]. Reversibility of the contraceptive effect and minimal side effects are two essential properties for a good oral male contraceptive. Without these properties, candidate drugs cannot be developed. This limitation may partially explain why the remaining compounds, including triptonide, have not been further explored for their male contraceptive potential. Given the well-documented sperm deformation effects induced by this herb, we decided to examine ten other compounds, aiming to identify one that can induce sperm deformation and male infertility without deleterious side effects. Among the compounds tested, we found that triptonide displayed almost ideal contraceptive effects in male mice, including excellent bioavailability allowing for oral administration, initial male infertility achieved by 3–4 weeks of daily oral treatment, sustained infertility in male mice maintained on the same dose for months, and reversal from infertile to fertile states in ~3–4 weeks after cessation of the treatment. Importantly, we observed no discernable deleterious effects, such as organ toxicity. Therefore, we proceeded and conducted the comprehensive proof-of-concept (POC) efficacy testing using triptonide on both mice and cynomolgus monkeys (*Macaca fascicularis*), and also attempted to identify its specific target(s) and mechanism(s) of action. Here, we report our data on the discovery of triptonide as an efficient, reversible male contractive agent in both mice and nonhuman primates. Based on the initial efficacy and safety data reported here, triptonide has the potential to become a promising nonhormonal male contractive agent.

## Results

**Oral administration of triptonide causes male infertility due to deformed sperm and reduced motility in mice.** As a pilot efficacy test, we first administered triptonide at five doses (0.1, 0.2, 0.4, 0.8, and 1.6 mg/kg BW) via oral gavage to adult male mice (C57BL/6J) at the age of 8–12 weeks. Mice from each dosing group were sacrificed weekly to examine both morphology and motility of epididymal sperm, as well as testicular histology for up to 6 weeks. We found the shortest time to achieve close to 100% sperm deformation characterized by head-bent-back and a complete lack of forward motility was between weeks 3 and 4 at the dose of 0.8 mg/kg BW or higher (Fig. 1a–e and Supplementary Movies 1 and 2). The pilot experiments were repeated two more times, and the same results were obtained. Thus, ~4 weeks of oral intake of triptonide at single daily doses of 0.8 mg/kg BW induce sperm deformation and loss of forward motility with close to 100% penetrance in adult male C57BL/6J mice. The "head-bent-back" phenotype in triptonide-treated sperm appeared to be homogenous and highly resembled the phenotype of *Spem1* KO sperm[5]. Transmission electron microscopic analyses confirmed that the sperm heads of triptonide-treated sperm were mostly bent at the connecting piece, and the bent head and neck were wrapped by residual cytoplasm (Fig. 1c). Moreover, scanning electron microscopic analyses revealed that testicular sperm already displayed the "head-bent-back" phenotype within the seminiferous epithelium, suggesting that the defects arise during spermatogenesis within the testis (Fig. 1c). In the official POC efficacy testing, after single oral daily doses of 0.8 mg/kg BW for 4 weeks, each of the treated male mice ($n = 12$) was mated with two fertility-proven female mice. While vaginal plugs were found in all of the females mated with the triptonide-treated male mice,

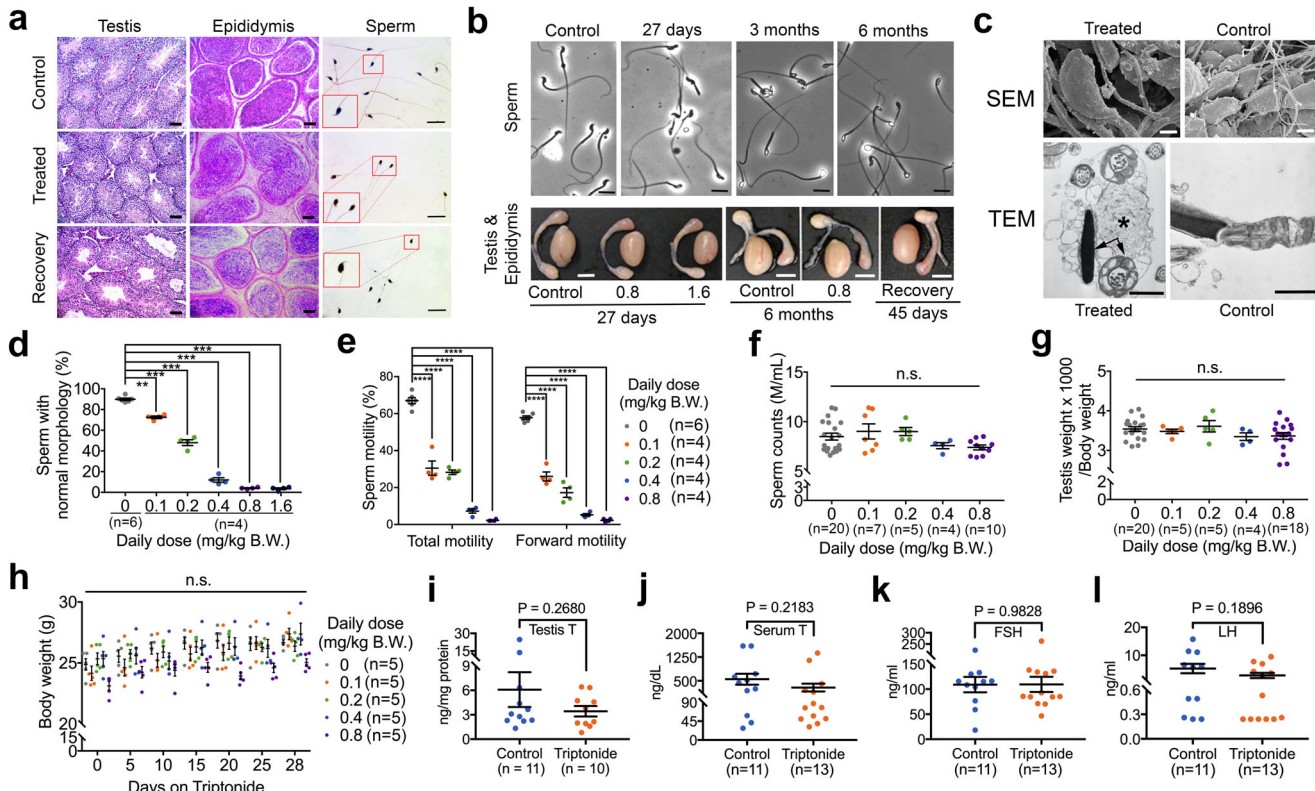

**Fig. 1 Proof-of-concept efficacy testing of male contraceptive effects of triptonide in mice. a** Representative histology of the testis, epididymis, and sperm from mice that received vehicle (control) and triptonide (treated) at single daily oral doses of 0.8 mg/kg BW for 4 weeks, and those recovered for 4 weeks after 4 weeks of triptonide treatment. Scale bars = 20 μm. Insets represent digitally enlarged framed areas. **b** Representative images of the gross morphology of sperm (upper panels, scale bars = 10 μm), testis, and epididymis (lower panels, scale bars = 20 mm) from mice that received short-term (27 days) and long-term (3 and 6 months) treatment with triptonide at single daily oral doses of 0.8 or 1.6 mg/kg BW, and those recovered for 45 days after 4 weeks of triptonide treatment. **c** Micrographs showing spermatozoa from triptonide-treated and control male mice. Bending of sperm heads occurs inside the seminiferous tubules of triptonide-treated male mice, as revealed by scanning electron microscopy (SEM; scale bar = 2 μm). Transmission electron microscopy (TEM) shows residual cytoplasmic contents (*) surrounding the bent sperm head and tail (arrows; scale bar = 1 μm). **d** Effects of different doses of triptonide (single daily oral gavage for 4 weeks) on sperm morphology in adult C57BL/6J male mice. Individual data points and mean (measure of center) ± SEM (error bars) are shown. **p < 0.01; ***p < 0.001, as determined by one-way ANOVA with dosing groups (n = 4) compared to the vehicle control group (n = 6). **e** Effects of different doses of triptonide (single daily oral gavage for 4 weeks) on total and forward sperm motility in adult C57BL/6J male mice. Individual data points and mean (measure of center) ± SEM (error bars) are shown. ****p < 0.0001, as determined by one-way ANOVA with dosing groups (n = 4) compared to the vehicle control group (n = 6). **f–h** Effects of different doses of triptonide (single daily oral gavage for 4 weeks) on sperm counts (**f**), relative testis weight (testis weight ×1000/body weight) (**g**), and body weight (**h**) in adult C57BL/6J male mice. Individual data points and mean (measure of center) ± SEM (error bars) are shown. Sample sizes are marked in brackets. No statistical significance (n.s.) among the groups based on one-way ANOVA analyses. **i–l** Effects of triptonide (single daily oral dose of 0.8 mg/kg BW for 4 weeks) on intratesticular testosterone (**i**), serum testosterone (**j**), FSH (**k**), and LH (**l**) levels in adult C57BL/6J male mice. Individual data points and mean (measure of center) ± SEM (error bars) are shown. Sample sizes are marked in brackets. No statistical significance was detected between the control and treated groups based on one-way analyses of variance (ANOVA) with p < 0.05 considered statistically significant.

none of the plugged females became pregnant. In contrast, female mice mated with vehicle control male mice (n = 12) were all pregnant. This was not surprising given that sperm in the treated mice were all deformed and displayed minimal or no forward motility (Fig. 1d, e). During the 4-week long triptonide treatment, no significant changes in either sperm counts, or testicular weight (Fig. 1f, g) were observed. Moreover, triptonide treatment did not change the body weight at any of the doses tested (Fig. 1h). To exclude the possibility that the male contraceptive effects were strain-dependent, we also tested triptonide on CD-1 mice by administering a daily oral dose of 0.8 mg/kg BW for 4 weeks. A similar phenotype was observed in the CD-1 mice (Supplementary Fig. 2).

At the end of the 4-week long triptonide treatment, we analyzed hormonal profiles and found no significant changes in the levels of FSH, LH, or testosterone (both serum and intratesticular; Fig. 1i–l). Consistent with the normal hormonal profiles, no gross or histological abnormalities were observed in the testis or epididymis (Fig. 1a, b). Either at the end of or during the treatment, male mice displayed frequent mounting behavior and successfully mated with adult female mice primed with PMSG and hCG, suggesting no adverse effects on mating behavior or capability. Examination of histology of major organs, including heart, liver, spleen, lung, and kidney revealed no pathological changes (Supplementary Fig. 3). Moreover, no discernable physical and behavioral abnormalities were observed among the treated mice. Taken together, these results suggest that a short-term oral intake of triptonide at a dose of 0.8 mg/kg BW causes no systematic toxic effects in major somatic organs in adult C57BL/6J male mice.

We also conducted long-term triptonide treatment using single daily oral doses of 0.8 mg/kg BW for 3 and 6 months, respectively

(Supplementary Movies 3-4). All 16 male mice tested (8 in each group) became infertile after 4 weeks, and infertility persisted between week 4 and the third or sixth month of treatment, as evidenced by the fact that none of the 32 fertile adult females placed into the individually housed, treated male mice (2 females per male) became pregnant. Further examination revealed a similar phenotype between long- (3 or 6 months) and short-term (4 weeks) treatment with triptonide (Fig. 1b). Similarly, no discernable toxic effects were observed in the mice that had undergone the long-term treatment of triptonide.

The sperm from triptonide-treated male mice cannot fertilize eggs either naturally or through in vitro fertilization due to deformation and a lack of forward motility. To test the fertility competence of sperm from triptonide-treated male mice, we conducted intracytoplasmic sperm injection (ICSI) by injecting the deformed sperm heads from triptonide-treated male mice (single daily oral doses at 0.8 mg/kg BW for 4 weeks) into WT MII oocytes. Interestingly, while the triptonide-treated sperm could fertilize the MII oocytes, the developmental potential of early embryos derived from the triptonide-treated sperm was significantly reduced compared to controls (Supplementary Table 1). Moreover, no pups were born after the embryos of 2-pronucleus (2PN) stage were transferred to recipient females, whereas a significant proportion (~15%) of the control 2PN embryos developed to full term and live pups were born in controls (Supplementary Table 2). These data suggest that triptonide-treated sperm may not be competent to support full-term development in mice.

**Oral administration of triptonide causes male infertility in cynomolgus monkeys.** To test whether triptonide would exert the same male contraceptive effects in primates, we also conducted POC efficacy testing using adult male cynomolgus monkeys. To determine the minimal effective dose, we first conducted a pilot study by treating four adult male cynomolgus monkeys with single daily oral administration of triptonide at the following four doses: 0.05, 0.1, 0.2, or 0.8 mg/kg BW. Semen samples were collected weekly, and sperm counts, motility, and morphology were analyzed to monitor the effects. We observed that single daily oral doses at 0.1 mg/kg BW or above resulted in oligo-astheno-teratozoospermia (OAT; Supplementary Fig. 4), characterized by sperm deformation (headless/tailless sperm, sperm with large cytoplasmic droplets, sperm with heavily coiled tails, etc.), minimal or no forward motility, and gradually reduced sperm counts after 5 weeks of triptonide treatment (Supplementary Movies 5 and 6). After a period of 3 months for drug "washout", these monkeys were retested using the same protocol, and the same results were obtained, i.e., single daily oral intake of triptonide (0.1 mg/kg BW) for 5 weeks caused OAT in adult male cynomolgus monkeys. Hence, the dose of 0.1 mg/kg BW was used for subsequent testing of short- and long-term male contraceptive efficacy of triptonide, as well as fertility reversibility.

In the short-term (8 weeks) POC efficacy testing, while vehicle-treated control monkeys displayed normal sperm counts, morphology, and motility (Fig. 2a–d), all of the seven adult male monkeys treated with triptonide showed deformed sperm (>95% deformation rate; Fig. 2a, b) with severely compromised forward motility (Fig. 2c; Supplementary Movies 7 and 8), and progressively reduced sperm counts (Fig. 2d). It is noteworthy that although the sperm counts in triptonide-treated male monkeys appeared to be reduced with time, the differences between treated and control groups did not reach statistical significance, probably due to great variations among samples collected at different timepoints and the limited number of monkeys analyzed. Consistently, testicular biopsies revealed that

elongating and elongated spermatids were largely absent, while other spermatogenic cells appeared to be normal in the seminiferous epithelium of treated monkey testes (Fig. 2e). As control, the testes from vehicle-treated monkeys showed robust spermatogenesis, with all types of spermatogenic cell present in the seminiferous epithelium (Fig. 2e). No significant decrease in testicular volume was observed between control and triptonide-treated groups (Supplementary Fig. 5). During the 8-week long treatment, weekly blood biochemical analyses revealed no major changes in hormonal (LH and testosterone) levels (Supplementary Fig. 6), blood chemistry (Supplementary Fig. 7), liver (Supplementary Fig. 8), or kidney (Supplementary Fig. 9) functions, suggesting that oral intake of triptonide at 0.1 mg/kg BW for 8 weeks causes male infertility without systemic side effects in adult male cynomolgus monkeys. We also continuously measured oxygen saturation for the first 7 weeks, and the readings ranged between 93–98%, and no differences were observed between treated and control groups (Supplementary Fig. 10).

In the long-term efficacy testing, four male monkeys were continuously treated with triptonide (at a daily oral dose of 0.1 mg/kg BW) for up to 126 weeks (~2.4 years), and no major health issues were observed except that all displayed OAT (Fig. 2f–h). On weeks 8 and 126, each of the control and treated males was paired with an adult female for 1 week to observe mating behavior, and the mating frequency appeared to be similar between the treated and vehicle-treated control groups (Fig. 2i and Supplementary Movie 9). To test fertility, two males from either control or treated group were paired with two fertility-proven females (one male with one female) between weeks 8 and 126. The two females paired with the two triptonide-treated males never became pregnant, whereas both of the females housed individually with the two control males did become pregnant. Of the two females impregnated by the two control males, one experienced a spontaneous miscarriage at ~3 months of gestation, and the other gave birth to a healthy male baby monkey at full term (~5.5 months) (Fig. 2j). These data suggest that the oral intake of triptonide at a daily dose of 0.1 mg/kg BW can cause male infertility due to OAT within 8 weeks, and that the effects can be potentially maintained indefinitely with oral administration of the same daily oral dose in adult male cynomolgus monkeys.

**Male infertility induced by triptonide is fully reversible in both mice and cynomolgus monkeys.** After 4 weeks of triptonide treatment with single daily oral doses of 0.8 mg/kg BW, all of the treated male mice displayed deformed sperm lacking forward motility with close to 100% penetrance. We then ceased triptonide treatment and added two to three adult fertility-proven female mice to each of four cages, where the treated male mice were individually housed (one treated male per cage). The first litter of pups were born within 40–45 days in each cage after cessation of triptonide treatment (Fig. 3a), suggesting that conception occurred between days 20 and 25 after cessation of triptonide treatment because the murine gestation length is ~20 days. Similarly, male fertility recovered within ~20–25 days following cessation of long-term (3 months) treatment of triptonide in male mice (Fig. 3a). Consistently, sperm morphology and motility returned to normal within 3 weeks after cessation of either short- (4 weeks) or long-term (3 months) triptonide treatment (Supplementary Movies 10 and 11). All pups sired by these fertility-recovered male mice were indistinguishable from those sired by control male mice, and both litter size and litter interval were comparable between control and treated males (Fig. 3b).

In POC efficacy testing using cynomolgus monkeys, four adult male monkeys that had received triptonide treatment for 10 weeks

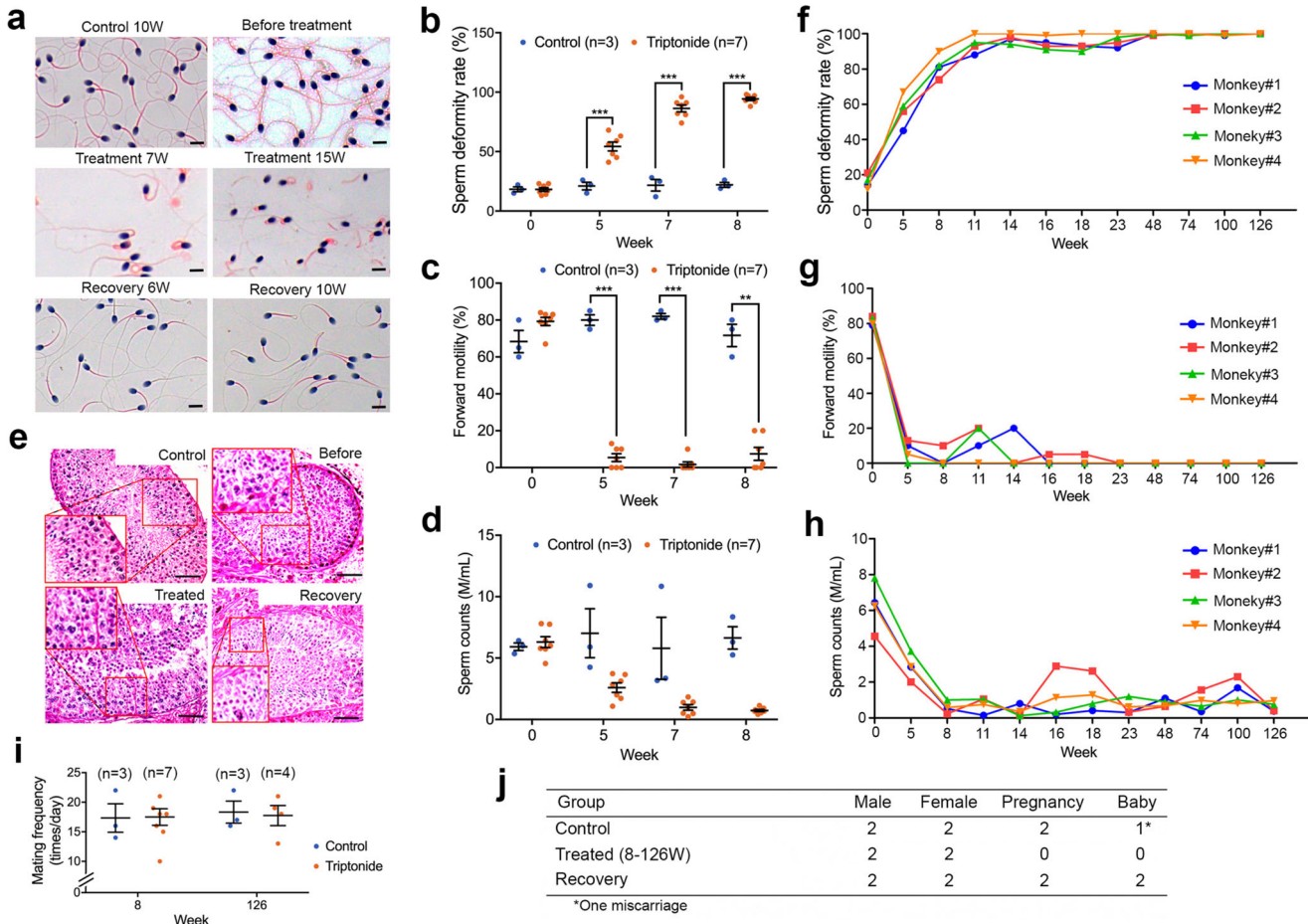

**Fig. 2 Proof-of-concept efficacy testing of male contraceptive effects of triptonide using cynomolgus monkeys. a** Representative images showing morphology of sperm from cynomolgus monkeys that received vehicle for 10 weeks (control), before and after triptonide treatment (single daily oral dose of 0.1 mg/kg BW for 7 and 15 weeks), or during recovery (6 and 10 weeks after 8 weeks of triptonide treatment). Scale bars = 5 μm. **b–d** Effects of short-term (8 weeks) triptonide treatment (single daily oral doses at 0.1 mg/kg BW) on the sperm morphology (**b**), sperm forward motility (**c**), and sperm counts (**d**) in adult male cynomolgus monkeys. Individual data points and mean (measure of center) ± SEM (error bars) are shown. \*\*$p < 0.01$; \*\*\*$p < 0.001$, as determined by two-way ANOVA with control ($n = 3$) compared to treated ($n = 7$) groups. **e** Representative testicular histology from monkeys that received vehicle (control), before and after triptonide treatment (single daily oral dose of 0.1 mg/kg BW for 8 weeks), or recovered from triptonide treatment (6 weeks after cessation of triptonide treatment). Insets represent digitally amplified, framed areas. Scale bars = 20 μm. **f–h** Effects of long-term (126 weeks) triptonide treatment (single daily oral doses at 0.1 mg/kg BW) on sperm morphology (**f**), sperm forward motility (**g**), and sperm counts (**h**) in 4 adult cynomolgus male monkeys. **i** Mating frequency of adult male monkeys that received either vehicle (control) or short-term (8 weeks) and long-term (126 weeks) triptonide treatment (single daily oral dose at 0.1 mg/kg BW). Individual data points and mean (measure of center) ± SEM (error bars) are shown. Sample sizes are marked in brackets. **j** Fertility performance of adult male monkeys that received vehicle (control) or triptonide (8–126 weeks), and those that recovered from 8 weeks of triptonide treatment (single daily oral dose at 0.1 mg/kg BW).

were transitioned to vehicle-only treatments between weeks 11 and 23, and their semen and blood samples were collected weekly to monitor the recovery process. Semen parameters among the recovery group reached control levels within 6 weeks after treatment cessation (Fig. 3c–e, and Supplementary Movies 12 and 13). Consistently, the monkeys recovered from triptonide treatment showed normal testicular histology (Fig. 2e) and testicular volume (Supplementary Fig. 5). During the 12-week long recovery period, weekly blood biochemical analyses revealed no pathological changes in hormonal (LH and testosterone) levels (Supplementary Fig. 6), blood cell counts (Supplementary Fig. 11), liver (Supplementary Fig. 12), or kidney (Supplementary Fig. 13) functions. The fertility-recovered males showed normal mating frequencies (Fig. 3f). Two of the four fertility-recovered male monkeys were individually paired with two fertility-proven adult females (one male with one female) for 6 months, and each of the female monkeys gave birth to a healthy full-term baby (Figs. 2j

and 3g). These results suggest that triptonide-induced male infertility is effective and reversible in primates.

### EC$_{50}$ analyses suggest that triptonide is a potent and safe male contraceptive agent.

To further evaluate the potency and safety of triptonide as a male contraceptive agent, we measured the 50% effective concentration (EC$_{50}$) of triptonide in inducing sperm motility loss and male infertility in adult C57BL/6J male mice. Based on the POC efficacy testing results (Fig. 1), adult C57BL/6J male mice of 8–12 weeks of age were treated with triptonide at 6 different doses (0, 0.1, 0.125, 0.2, 0.4, and 0.8 mg/kg BW, p.o. daily) for 4 weeks. At the end of week 4, sperm motility and morphology, as well as fertility were analyzed. By plotting sperm motility against each dose tested, values of the EC$_{50}$ for inhibiting total and forward motility were determined to be 0.11 and 0.10 mg/kg BW, respectively (Fig. 4a). The most meaningful EC$_{50}$

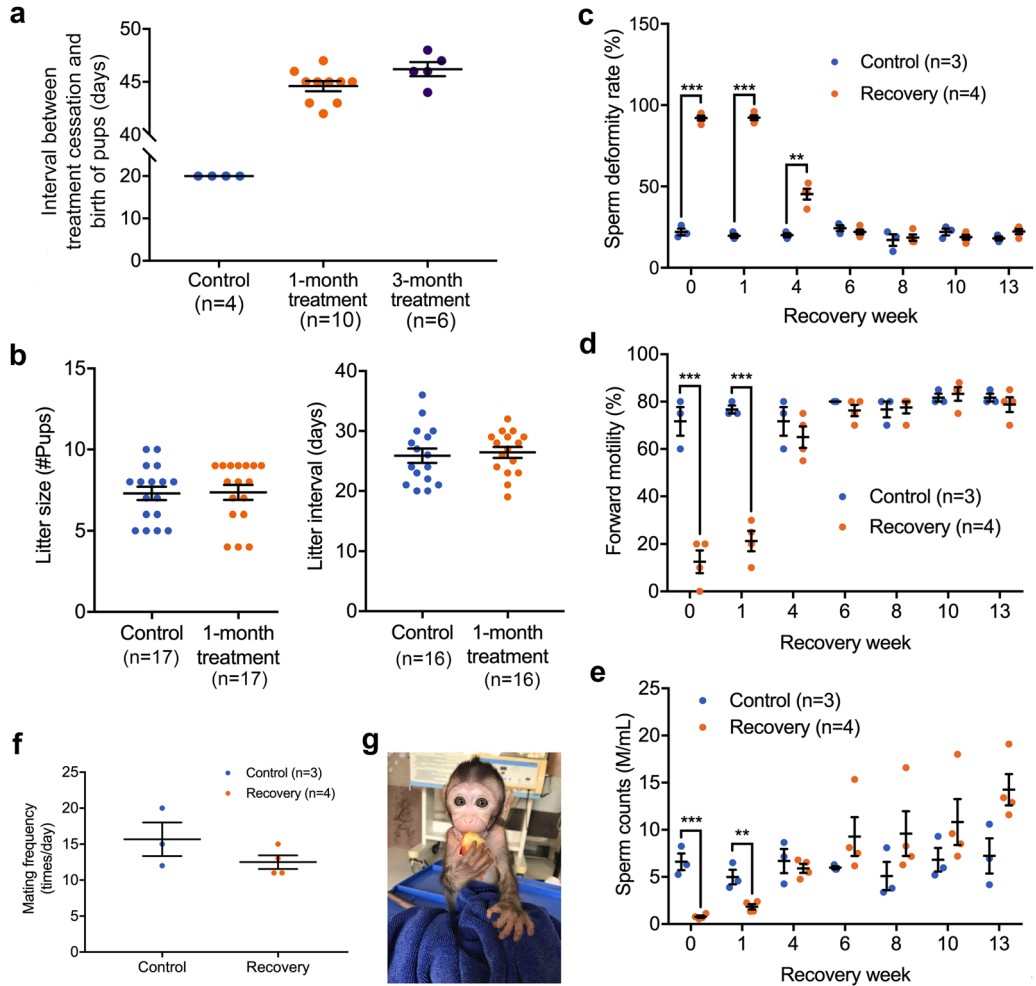

**Fig. 3 Reversibility of triptonide-induced male infertility in adult mice and cynomolgus monkeys. a** Male mice treated with triptonide (single daily oral doses at 0.8 mg/kg BW) for 4 weeks regained their fertility between days 20 and 25 after cessation of the treatment, as demonstrated by the birth of pups by females mated with previously treated males (single daily oral doses of 0.8 mg/kg BW for 4 weeks). Individual data points and mean (measure of center) ± SEM (error bars) are shown. Sample sizes are marked in brackets. **b** Litter size and interval of fertility-proven females mated with control male mice or those recovered from 1-month triptonide treatment (single daily oral dose at 0.8 mg/kg BW). Individual data points and mean (measure of center) ± SEM (error bars) are shown. Sample sizes are marked in brackets. No statistical significance between the control and treated groups based on the Kolmogorov–Smirnov *t* test. **c**–**e** Recovery of sperm morphology (**c**), sperm forward motility (**d**), and sperm counts (**e**) in adult male cynomolgus monkeys after 8 weeks of triptonide treatment (a single daily oral dose at 0.1 mg/kg BW). Individual data points and mean (measure of center) ± SEM (error bars) are shown. **p < 0.05; ***p < 0.01, two-way ANOVA with the recovery group (n = 4) compared with the control group (n = 3). The exact p values can be found in the Source data file. **f** Mating frequency of adult male monkeys that recovered for 8 weeks from triptonide treatment (single daily oral dose at 0.1 mg/kg BW for 8 weeks). Individual data points and mean (measure of center) ± SEM (error bars) are shown. Sample sizes are marked in brackets. **g** A male baby monkey fathered by an adult male monkey recovered from 8 weeks of triptonide treatment (a single daily oral dose at 0.1 mg/kg BW).

value should be based on the male fertility. To this end, 14 adult C57BL/6J male mice of 8–12 weeks of age were treated with triptonide at five different doses (0, 0.1, 0.2, 0.4, and 0.8 mg/kg BW, p.o. daily) for 4 weeks. After 4 weeks of triptonide treatment, each male was mated with two to three adult fertility-proven female mice in three independent experiments, and pregnancy rates were calculated based on percentage of pregnancies among all of the plugged females (Supplementary Table 3). The $EC_{50}$ of triptonide in causing male infertility was determined to be at 0.09 mg/kg BW (Fig. 4b). The $LD_{50}$ of triptonide (p.o.) was previously shown to be 300 mg/kg BW in mice, 980 mg/kg BW in rats, and 3200 mg/kg BW in rabbits (Material Safety Data Sheet from the Clearsynth Labs Pvt. Ltd; Patent International Publication#: WO/2016/205539; International Application #: OCT/US2016/037900). Thus, the $EC_{50}$ (~0.1 mg/kg BW) of triptonide is ~1/8 of the minimal effective dose, and ~1/3000 of the $LD_{50}$ in mice,

highlighting the excellent potency and safety of triptonide as an oral male contraceptive agent.

**Triptonide does not cause DNA double-strand breaks**. A previous report noted that triptonide may possess DNA alkylating activity due to its epoxide structure[25]. If triptonide alkylates DNA, it would cause DNA double-strand breaks (DSBs), which can be marked by γH2AX (the phosphorylation of the histone H2AX on serine 139)[26], a well-known indicator for DNA DSBs[27,28]. To this end, we performed immunohistochemistry and western blots to detect γH2AX levels in triptonide-treated mouse testes (Supplementary Fig. 14). To investigate whether triptonide causes DNA DSBs in somatic tissues, we also examined levels of γH2AX in the liver of triptonide-treated and control males (Supplementary Fig. 14). No significant changes in γH2AX levels were observed in either the testis or liver, suggesting that

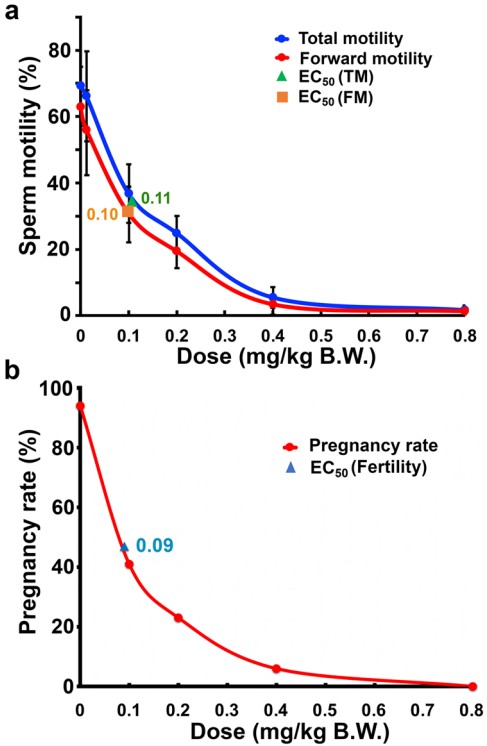

**Fig. 4 The EC$_{50}$ of triptonide in adult male C57BL/6J mice. a** The EC$_{50}$ of triptonide in inhibiting sperm total (TM) and forward (FM) motility in adult male mice. Data points represent mean ± SD; $n = 5$ for doses 0, 0.4, and 0.8 mg/kg BW, and $n = 9$ for doses 0.1, 0.125, and 0.2 mg/kg BW. **b** The EC$_{50}$ of triptonide in inducing male infertility in adult mice. Data points represent pregnancy rate, which reflects percentage of pregnancy among all of the plugged females by the males treated with five doses of triptonide (0, 0.1, 0.2, 0.4, and 0.8 mg/kg BW) for 4 weeks. $n = 14$ for each of the five doses tested.

triptonide does not cause DNA DSBs. Failed repair of DNA DSBs is known to cause male germ cell depletion through apoptosis, and massive apoptosis would cause disrupted spermatogenesis characterized by multinucleated cells and sloughing of the damaged spermatogenic cells into the lumen of the seminiferous tubules[29–31]. None of these effects was observed in triptonide-treated testes, which are, in fact, largely indistinguishable from control testes (Fig. 1a). Similarly, no compromised liver functions were detected in the liver panel of blood tests (Supplementary Figs. 8 and 12). Moreover, these findings are also consistent with the fact that neither tumorigenesis nor any other deleterious side effects were observed in any of the triptonide-treated mice or monkeys that received long-term triptonide treatment (6 months for mice and >2.4 years for cynomolgus monkeys).

**Identification of junction plakoglobin/gamma-catenin as the candidate target of triptonide**. To identify the drug target that mediates the effect of triptonide treatment, we adopted the drug affinity responsive target stability (DARTS) assay, which represents an unbiased approach for the identification of drug targets[32,33]. DARTS is based upon the fact that when a compound binds specifically to its protein target, the presence of the molecule protects a part of the target protein from being hydrolyzed by proteases (Fig. 5a)[32,33]. We first performed in vitro DARTS experiments, where testis lysates were incubated with triptonide or vehicle (0.01% DMSO) followed by digestion, with either pronase or thermolysin at different concentrations for variable lengths of incubation. But we observed no protected protein

bands after polyacrylamide gel electrophoresis (PAGE). We then investigated an in vivo variation of DARTS by extracting protein from testis lysates from mice treated with 0.8 mg/kg BW triptonide or vehicle (control) for 4 weeks. The precipitated protein was digested with trypsin for various lengths of time followed by PAGE. On the PAGE gels, one band of ~18 kDa was consistently seen in the treated, but not in the control, lysates in two independent experiments (Fig. 5b). Mass spectrometry (MS) of the band consistently identified four proteins, including keratin 5, junction plakoglobin (JUP; also called gamma-catenin), tubulin β-4B, and polyubiquitin C. Given that keratins are the most common contaminants in MS-based proteomic analyses, other proteins remained the more probable candidate targets.

We then attempted a GlycoLink resin microcolumn-based affinity purification method (Fig. 5c). The columns are intended for attachment of sugar molecules after the *cis* diols are oxidized to produce aldehydes, but they also react with ketones via a Schiff base reaction. The attachment reaction was achieved using an acidic buffer or an optional 0.2 M carbonate/bicarbonate basic (pH = 9.4) buffer. Acidic conditions could facilitate attachment of triptonide to the beads via the ketone or possibly the lactone, leaving unhindered epoxide groups, whereas basic conditions may promote binding via one of the epoxide groups. We, therefore, used both acidic and basic buffers to promote attachment of triptonide to the beads. A ~60 kDa band was detected in both triptonide and control conditions from both acid-conjugated and base-conjugated beads (Fig. 5d), but the bands were much lighter in controls than in triptonide-bound beads (Fig. 5d), suggesting that the binding might be specific. The bands were cut out and subjected to MS analyses which identified ~30 proteins, among which three proteins were also identified by the DARTS assays, including keratin 5, tubulin β-4B, and junction plakoglobin/gamma-catenin. To further validate these results, we attached triptonide to GlycoLink resin microcolumns using acidic buffer. Control resin was prepared with no attached triptonide. Total testicular proteins were then incubated with the triptonide-attached and control resin. After washing, the eluates were subjected to western blot analyses using anti-keratin 5 and anti-junction plakoglobin antibodies. Junction plakoglobin/gamma-catenin was detected (Fig. 5e), whereas keratin 5 appeared to be negative (Fig. 5f), suggesting that the binding between triptonide and junction plakoglobin/gamma-catenin is specific.

Given that triptonide-treated sperm phenocopy *Spem1*-null sperm[5], characterized by deformed sperm (head-bent-back) with no or minimal motility, we also adopted a candidate approach to explore whether SPEM1 interacts with any of the proteins identified to bind triptonide. Protein structural analyses predict that the N-terminus of SPEM1 acts as a domain for protein–protein interactions. Thus, the N-terminal portion of SPEM1 (~28 a.a.) was synthesized and purified. A biotin was added onto the C-terminus of this peptide. The biotin-labeled N-terminal 28 a.a. peptides of SPEM1 were attached to streptavidin beads, which were then used to probe testicular lysates prepared from an 8-week-old male mouse (Fig. 5g). Using this method, five proteins were identified to bind only to the N-terminus peptides, but not the control beads, including junction plakoglobin/gamma-catenin, heat shock-related 70 kDa protein 2 (HSP70-2), L-lactate dehydrogenase C chain (LCDHC), Y-box-binding protein 2, and retinal dehydrogenase 1. Interestingly, Y-box protein 2, LCDHC, and retinal dehydrogenase 1 were also identified to bind triptonide in the GlycoLink bead-based affinity purification assays (Fig. 5h). Taken together, these observations indicate that the SPEM1 N-terminus appears to interact with a number of proteins, including junction plakoglobin/gamma-catenin. Given that junction plakoglobin/gamma-catenin was

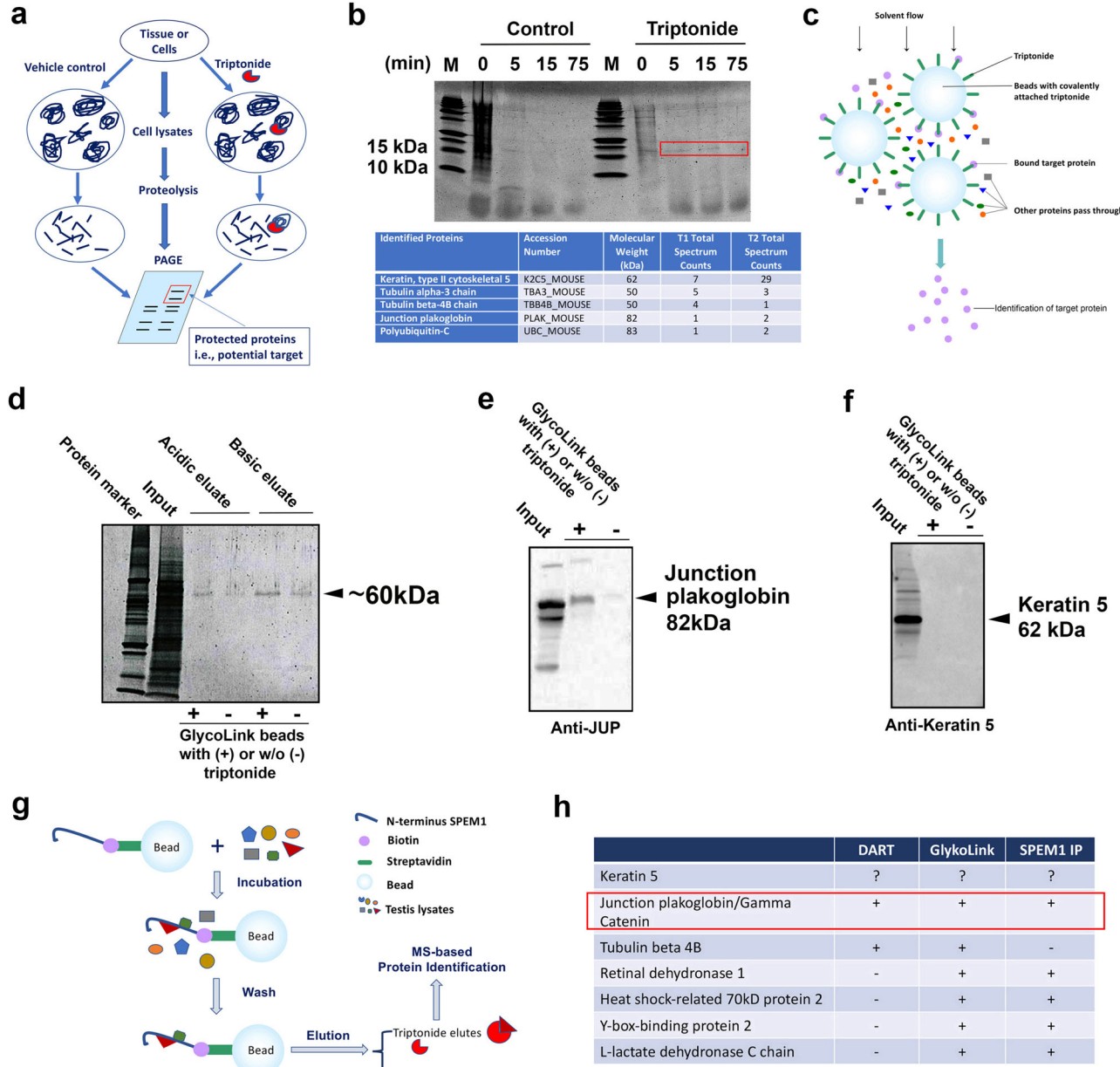

**Fig. 5 Identification of potential target(s) of triptonide in murine testes. a** Schematics showing a modified drug affinity responsive target stability (DARTS) assay used in this study to identify proteins interacting with triptonide. Protein lysates from total testes of adult male mice treated with triptonide (single daily oral dose at 0.8 mg/kg BW for 4 weeks) were subjected to digestion using different proteinases for various durations. **b** A representative gel image (upper panel) showing a specific band of ~18 kDa (red frame) unique to the triptonide-treated testes. The bands were cut out for mass spectrometry-based protein identification, and the top five hits were listed in the table (lower panel). **c** Schematics showing the GlycoLink beads-based affinity purification method used in this study. **d** A representative gel image showing a specific band of ~60 kDa eluted from triptonide-conjugated beads under acidic and basic conditions. The "input" lane shows the amount of testis lysates used in the assays. The bands were subjected to MS-based protein identification and some of the proteins identified are listed in **h**. **e** A representative western blot showing detection of junction plakoglobin/gamma-catenin in the eluates from beads conjugated with or without triptonide, as all we as the input of total testicular lysates used in the assays. **f** A representative western blot showing detection of keratin 5 in the eluates from beads conjugated with or without triptonide, as all we as the input of total testicular lysates used in the assays. **g** Schematics showing the immunoprecipitation-based identification of proteins interreacting with the N-terminus of SPEM1. The biotin–streptavidin system was utilized to bind the N-terminus of SPEM1 to magnetic beads. The major proteins identified using this method are listed in **h**. **h** A summary of major proteins identified by the three methods used in this study. Junction plakoglobin/gamma-catenin appears to be the most likely drug target (red frame).

also detected to bind triptonide in both DARTS and GlycoLink bead-based affinity purification assays, these data strongly suggest that junction plakoglobin/gamma-catenin is the most likely functional target of triptonide. Thus, triptonide binding of junction plakoglobin/gamma-catenin may interfere with its interactions with SPEM1 (Fig. 6), leading to defects in cytoplasmic removal and consequently bending of sperm heads, a phenotype resembling *Spem1*-null sperm[5].

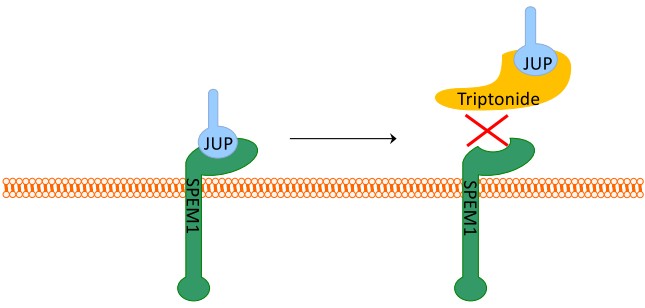

**Fig. 6 The proposed action of triptonide in inducing sperm deformation and male infertility.** Physiologically, SPEM1 interacts with junction plakoglobin through its N-terminus. However, triptonide binds junction plakoglobin with higher affinity, and this binding disrupts normal interactions between junction plakoglobin with SPEM1, causing a phenotype similar to that of *Spem1* knockout, i.e., sperm deformation and male infertility.

## Discussion

The standard drug development process often starts with the identification of a drug target using either in vivo (e.g., genetic models) or in vitro (e.g., cell culture-based screening) assays, and proceeds with high-throughput screening of millions of compounds to identify those that interact with the targets, followed by animal testing for efficacy and safety. However, it must be emphasized that serendipity also plays a critical role in drug discovery[34–36]. For example, ~20% of the pharmaceuticals in clinical use today are chemical derivatives of drugs discovered serendipitously[34]. The so-called "antisperm" effects of the Chinese herb (*T. wilfordii Hook F*, commonly known as lei gong teng or thunder god vine) was serendipitously discovered because physicians noticed that some of the men who took this Chinese herbal medicine for an extended period (>3 months) displayed compromised fertility[12]. Although this observation triggered a wave of intensive research on the potential use of this herb or compounds isolated form this herb as male contraceptive agents, the enthusiasm soon dwindled because several initial studies found that either the crude herbal extracts or several of the most abundant compounds purified from this herb could indeed cause male infertility, but all exerted severe toxic side effects and infertility could hardly be reversed after the treatment was stopped[12–18,20–24].

Based on our genetic study on murine spermiogenesis using gene KO technologies, we put forward the idea for developing nonhormonal male contraceptives through disrupting the last several steps (i.e., post elongation) of spermiogenesis. The idea was based upon the fact that genetic disruptions of mid- or late spermiogenesis (from elongation onward) rarely cause apoptosis or massive germ cell depletion within the seminiferous epithelium, although sperm are consistently deformed and/or display no or very limited motility[4]. For example, *Spem1* encodes a protein exclusively expressed in elongating and elongated spermatids, and ablation of *Spem1* causes male infertility due to sperm deformation characterized by the head-bent-back by 180 degrees, and the bent head and neck surrounded by residual cytoplasm[5]. SPEM1 is mainly localized on the manchette, which is a microtubular apparatus unique to elongating and elongated spermatids, and is believed to function to facilitate nucleocytoplasmic transport[5]. In the absence of SPEM1, the coordinated nuclear and cytoplasmic elongation processes become unsynchronized with nuclear elongation continuing, while cytoplasmic elongation ceasing, leading to head bending followed by the tail wrapping around the bent head inside the residual cytoplasm[4]. In the absence of depletion of defective late spermatids within the

seminiferous epithelium, testicular histology of *Spem1* KO mice is indistinguishable from that of wild-type controls except that all of the sperm are deformed and lack forward motility. Based on the phenotype of *Spem1* KO male mice, it is conceivable that if a compound directly or indirectly targets SPEM1, a phenotype similar to *Spem1* KO sperm would be expected. Indeed, our data suggest that triptonide appears to be such a compound. Triptonide displays excellent bioavailability allowing for oral administration. It induces total infertility in weeks (3–4 weeks in mice and 5–6 weeks in monkeys) and maintains the infertile state for months and years with the same dosage. Male fertility can be regained ~4–6 weeks after cessation of triptonide treatment. More importantly, no discernable deleterious side effects, e.g., organ toxicity, have ever been observed in our efficacy testing on either mice or monkeys. The discovery of triptonide as a promising male contraceptive agent serves as a testimony to the power of our strategy in developing male contraceptives.

Oral intake of triptonide requires 3–4 and 5–6 weeks to achieve full male infertility in mice and monkeys, respectively. The latent effect implies that triptonide acts on a specific step during spermiogenesis. Indeed, earlier studies have estimated that the duration of spermatogenic cycles is ~35 days in mice and ~42 days in cynomolgus monkeys[37,38]. Therefore, the one-week difference in the induction of full male infertility effects between mice and monkeys likely reflects the difference in the duration of spermatogenic cycle between these two species. It is noteworthy that the decrease in sperm counts induced by triptonide was more significant in cynomolgus monkeys than in mice, which may reflect differences in spermatogenesis between these two species. Indeed, the testicular histology is indistinguishable between triptonide-treated and control mice, whereas triptonide-treated monkey testes lacked elongating and elongated spermatids, suggesting that while triptonide may target the same protein within the testes in both species, the biological consequences may differ. Our drug target analyses strongly suggest that triptonide most likely binds junction plakoglobin/gamma-catenin and thus, disrupts its interactions with SPEM1, leading to a phenotype similar to that observed in *Spem1* KO testes. Phylogenetic analyses have shown that mouse and monkey SPEM1 proteins share ~69% amino acid sequence similarity[5], suggesting SPEM1 function has evolved from mice to monkeys, and this may explain the phenotypic differences observed between triptonide-treated mice and cynomolgus monkeys. Of interest, levels of major hormones, including FSH, LH, and testosterone, in treated mice or monkeys did not vary significantly from those of controls, suggesting that triptonide treatment does not adversely affect libido or other endocrine functions. More importantly, once triptonide treatment is stopped, infertile male mice and monkeys regain their fertility in ~3–4 and ~5–6 weeks, respectively. Thus, the recovery timeline is similar to that required to initially induce infertility, further supporting the notion that triptonide acts on one or more specific steps during spermiogenesis.

The biggest concern associated with the development of any contraceptives is the potential deleterious side effects. This was particularly true for our study because several of the major compounds isolated from *T. wilfordii* have been shown to exert significant toxic effects[12–18,20–24]. For treatment of life-threatening diseases, such as cancer, a certain level of deleterious side effects can be viewed as an acceptable consequence if unavoidable. However, for the development of a male contraceptive, even minor side effects may be sufficient to discourage its use by healthy men who are active both reproductively and sexually. Therefore, our finding that neither short- (4 weeks for mice and 8–24 weeks for monkeys) nor long- (3 or 6 months for mice and up to 2.4 years for monkeys) term oral intake of triptonide causes discernable toxic side effects in either species

represents a major asset in support of the potential advancement of this compound to the investigational new drug status and to the subsequent clinical trials.

A previous report suggested that triptonide may have DNA alkylating activity due to its epoxide structure[25]. However, for epoxides to alkylate DNA, an aromatic ring is required to intercalate between base pairs in the double helix so that the epoxide ring can react with a nitrogen atom in a DNA base[39]. Triptolide is structurally very similar to triptonide, and none of the multiple anticancer mechanisms of triptolide involves alkylation of DNA[40] given that triptonide is too large to intercalate between DNA base pairs[41]. The lack of alkylating activity of triptonide is further supported by the following facts: (1) this herbal medicine has been used by millions of people in China for many decades to treat a wide variety of dermatological conditions with no reports of increased tumorigenesis among long-term users. (2) None of the mice or monkeys that underwent long-term (3–6 months for mice and up to 2.4 years for monkeys) oral triptonide treatment in our POC efficacy tests developed cancer. (3) DNA alkylating activity tends to cause DNA DSBs, but no increase in gamma-H2AX levels was detected in triptonide-treated testes.

Another line of evidence supporting the safety of triptonide comes from the fact that its $EC_{50}$ associated with induction of infertility and sperm motility loss is ~0.1 mg/kg BW, which is ~1/8 of the minimal effective dose (0.8 mg/kg) and ~1/3000 of $LD_{50}$ (300 mg/kg BW) in mice. In other words, triptonide is very potent in suppressing male fertility because even 1/8 of the normal dose will generate maximum efficacy in half of the mice treated. Thus, the effective dose of triptonide treatment required to achieve male contraception is far below the level that would induce any sort of deleterious side effects.

Given that triptolide is the most abundant compound in the Chinese herb *T. wilfordii*, triptonide purified from this herb can sometimes be contaminated with trace amounts of triptolide or other abundant compounds, which, unlike triptonide, have been shown to be highly toxic[42]. Therefore, trace contaminants would cause side effects, which are, in fact, not due to triptonide. Indeed, one of the batches of triptonide we purchased included a stated purity of 96%, but our testing on mice revealed multiple side effects similar to those seen in mice treated with triptolide. Therefore, it is critical to use highly pure triptonide for POC efficacy testing. We sent the two batches of purified triptonide and one batch of chemically synthesized triptonide used in this study to an independent commercial lab for purity tests, using both NMR and elemental analyses (Supplementary Fig. 15), and the results showed that all three were nearly 100% pure. Taking into consideration the stated margin of error of these testing systems (~0.4%), we determined that the purity of triptonide used in our study was at least 99.6%. We used the synthetic triptonide and performed POC efficacy testing on mice, and the same reversible male contraceptive effects were obtained (Supplementary Fig. 16). These data suggest that the male contraceptive effects of triptonide reported here are indeed caused by triptonide, rather than the potential trace contaminants in triptonide used.

It is noteworthy that the main focus of the present study was POC efficacy testing, and a lack of discernible side effects was based upon the observation of behavior (for both mice and monkeys), examination of gross morphology, and histology of vital organs (for mice only), as well as hematological and serum biochemical analyses (for monkeys only). While the data reported here suggest triptonide might be a promising contraceptive drug candidate for men, its safety needs to be further evaluated through comprehensive pharmacokinetics and toxicology studies in the future. An intriguing question is why triptonide only exerts effects on late spermiogenesis without affecting functions of other major organs? One explanation is that it may target a protein that is exclusively expressed in elongating and elongated spermatids. However, our drug target identification assays point to junction plakoglobin/gamma-catenin as its most likely target, which is known to be expressed ubiquitously[43,44]. However, SPEM1, which is an interacting partner of junction plakoglobin/gamma-catenin, is indeed exclusively expressed in elongating and elongated spermatids during spermiogenesis[5]. Therefore, it is highly likely that proper function of SPEM1 requires normal interactions with junction plakoglobin/gamma-catenin, and SEPM1 function is abolished once interactions between SPEM1 and junction plakoglobin/gamma-catenin are disrupted in the presence of triptonide. Although interactions between junction plakoglobin/gamma-catenin and SPEM1 may explain the testis-specific effect of triptonide, such a mechanism needs further validation using genetic models and in vitro assays in the future. Furthermore, it is critical to understand how triptonide is metabolized once it gets into the bloodstream and major organs, including the testis. To this end, a thorough pharmacokinetics study is warranted. To establish triptonide as an ultimate drug, scalable, efficient, and economic chemical synthesis of triptonide is also critical. If SPEM1 is proven to be the true target of triptonide, other small compounds that can specifically interact with SPEM1 may exert a similar male contraceptive effect to triptonide with even greater specificity and efficacy.

In summary, we report the discovery of triptonide as a potent and safe nonhormonal male contraceptive agent in mice and nonhuman primates. The success also validates a more general idea for developing male contraceptives. i.e., disabling, rather than "killing"/depleting, all sperm by targeting proteins or other gene products specific to the elongation steps during spermiogenesis. Finally, we note that serendipitous drug discovery should continue to be encouraged and supported as an integral part of the ongoing efforts to develop nonhormonal male contraceptives.

## Methods

**Animal use and care**. Male and female adult (2–3 months) mice of C57BL/6J and CD-1 strains were used. Mice were housed in a specific pathogen-free and temperature- and humidity-controlled facility under a light–dark cycle (10 h light and 14 h dark) with food and water ad libitum at the University of Nevada, Reno. Animal use protocol was approved by Institutional Animal Care and Use Committee of the University of Nevada, Reno, and is in accordance with the "Guide for the Care and Use of Experimental Animals" established by National Institutes of Health (1996, revised 2011). All cynomolgus monkeys used were housed at the Blooming Spring Biological Technology Development Co. LTD, in Guangzhou, China, which is fully accredited by the Association for the Assessment and Accreditations of Laboratory Animal Care International. The animal use protocol was approved by the Research Ethics Committee of the Blooming Spring Biological Technology Development Co. LTD. A total of 12 healthy male adult (9–13 years of age with body weight ranging between 4.96 and 11.80 kg) and 6 fertility-proven adult females (9–13 years of age with body weight ranging between 3.89 and 4.00 kg) cynomolgus monkeys were used in this study. The general information of the monkeys used in this study is summarized in Supplementary Table 4.

**Chemicals**. Triptonide ($C_{20}H_{22}O_6$, MW: 356.39 g/mol) was purchased from the Chengdu Biopurity Phytochemicals (purity > 98%, Lot#: 15033012 and 14081502) and MCE (MedChem Express; purity > 98%, Cas#: 38647-11-9. Lot#13216). DMSO was purchased from Sigma-Aldrich. In addition to HPLC data provided by the company, the purity of triptonide used was independently validated using NMR elemental analyses in NuMega Resonance Labs, Inc. (San Diego, CA; Supplementary Fig. 15).

**Proof-of-concept efficacy testing in mice**. For POC efficacy testing on mice, a stock solution of triptonide was prepared at 4 mg/ml in DMSO. Each vial was wrapped with aluminum foil to block light and stored at −20 °C. For gavage feeding, 125 μl of triptonide working solution (5 μl stock solution was added into 120 μl of PBS) containing 20 μg triptonide were administered to a mouse of 25 g (0.8 mg/kg B.W) using a gavage feeding needle (20 G 1–1/2 in., Cadence Science, Japan). Other doses (0.1, 0.2, 0.4, and 1.6 mg/kg BW) were prepared accordingly. Control mice received vehicle (4% DMSO in PBS). After oral administration of triptonide or vehicle, the mice were observed every hour for the first 3 h, followed by daily monitoring thereafter.

In the initial pilot study to identify minimal effective dose and duration, five doses (0.1, 0.2, 0.4, 0.8, and 1.6 mg/kg BW) were tested using 18 mice per dose group. Three mice from each dose group were sacrificed weekly to examine sperm parameters and testicular histology for up to 6 weeks. In the subsequent official POC efficacy testing, 6–20 mice in each test group were treated with triptonide at one of the four doses (0.1, 0.2, 0.4, and 0.8 mg/kg BW) via single daily oral gavage. At the end of the fourth week, all triptonide-treated and control mice were sacrificed and caudal epididymal sperm parameters were measured using a computer-assisted sperm analysis (CASA) system (Sperm Analyzer Mouse Traxx, Hamilton-Thorne). In addition, blood samples were collected and allowed to clot at room temperature for 1.5 h. After disrupting clot adhesion to the tube wall, the blood samples were centrifuged at $2000 \times g$ for 10 min at room temperature. Serum was collected into a polypropylene microcentrifuge tube and stored at $-20\,°C$ for hormonal measurements. Hormonal levels were measured at the Ligand Assay and Analysis Core, Center for Research in Reproduction, University of Virginia School of Medicine, Charlottesville, Virginia. The Mouse Pituitary Magnetic Bead Panel Multiplex Kit (Cat.# MPTMAG-49K; Lot#3141797; Millipore Corporation, Billerica, MA) was used to determine the levels of FHS and LH following the manufacturer's instructions. The reportable range the FSH/LH assays was between 0.48 and 300 ng/ml, and the intra-assay CV was between 0.4 and 2.9% for FSH, and 3 and 7.8% for LH. To measure intratesticular testosterone levels, testicular homogenates were prepared by homogenizing the testis in cold $(4\,°C)$ PBS (without detergent or EDTA) on ice for 60 s followed by centrifugation $(2000 \times g$ for 10 min at $4\,°C)$ to remove cell debris. The supernatants were then collected into 1.5 ml Eppendorf tubes and stored at $-80\,°C$. Both serum and testicular testosterone levels were assayed using the Testosterone Mouse & Rat ELISA Kit (Cat. #IB79106; Kit Lot# 28K088-2; Immuno-Biological Laboratories, Inc., Minneapolis, MN) following the manufacturer's instructions. The assay is a solid phase enzyme-linked immunosorbent assay based on the principle of competitive binding. The microwells coated with a monoclonal antibody against a unique antigenic site of testosterone were first incubated with the samples so that testosterone in the samples competes with testosterone-conjugated horseradish peroxidase. After washing to remove the unbound molecules, the solid phase was then incubated with the substrate and the colorimetric reaction was used to quantify the levels of testosterone in the samples. A standard curve was first constructed by plotting OD values against concentrations of the standards, and the concentration of unknown samples were determined using this standard curve. The reportable range of this assay was between 8.3 and 1600 ng/dl, and intra-assay CV was between 0.8 and 3.7%.

Histology of all major organs including heart, liver, spleen, lung, lung, brain, colon, small intestine, testis, and male reproductive tracts (epididymis, seminal vesicle, and prostate) was analyzed in-house. Testes were fixed in Bouin's fixatives and other organs were fixed in 10% neutral buffered formalin followed by embedding into paraffin. Sections were cut followed stained with hematoxylin and eosin for microscopic evaluation.

To test fertility, after 3–4 weeks of oral administrations of triptonide and vehicle control, two or three females were added to individual cages containing single triptonide-treated or control male mice. The female mice were examined every morning for the presence/absence of vaginal plugs. Once plugs were identified, the female mice were removed and after 7 days, they were transferred to a new cage, and the pregnancies and pups were observed and recorded.

**Electron microscopy of mouse seminiferous tubules and sperm.** Both scanning and transmission electron microscopic analyses were conducted as described[5]. Scanning electron microscopy was conducted using stages VII and VIII semi-niferous tubules dissected from control and triptonide-treated testes, whereas TEM was performed on sperm collected from the cauda epididymis of control and triptonide-treated male mice.

**Intracytoplasmic sperm injection.** Mouse ICSI was performed as described[5]. Cauda epididymal sperm were collected from control and triptonide-treated male mice. Only sperm heads were injected into MII oocytes from donor female mice. To study developmental potential of the preimplantation embryos, the injected oocytes were cultured in vitro. For evaluation of full-term development, the 2-pronuclear embryos were transferred into the recipient females.

**Proof-of-concept efficacy testing in cynomolgus monkeys.** To identify minimal effective dose and duration, four cynomolgus monkeys were treated with triptonide at four doses (0.05, 0.1, 0.2, 0.8, and 5 mg/kg BW) daily. Based on the weight of each monkey, the total amount of triptonide equivalent to 40 days of oral treatment of 0.1 mg/kg BW was calculated and weighted for making the stock solution in 2 ml DMSO. The working solution was prepared by diluting the stock solution with 18 ml PBS followed by aliquoting into 40 vials (0.5 ml each) for storage at $-20\,°C$. One vial was thawed each day and the triptonide working solution (0.5 ml) was added into the food (steamed bun, bread, apple, banana, etc.) before feeding. The technician observed the feeding to make sure the triptonide-containing food was completely eaten. By the end of each week, semen and blood samples were collected for analyses of sperm parameters and blood chemistry. The pilot test was stopped at week 9 because almost all sperm became deformed with minimal or no forward motility and sperm counts had been drastically reduced. One monkey that received 5 mg/kg BW triptonide daily p.o. displayed signs of side effects starting at week 4 and became worse thereafter; thus, the experiment was stopped at the end of week 5 and that monkey was not used in subsequent POC efficacy testing.

Based on the pilot test, the minimal effective dose was determined to be 0.1 mg/kg (BW). In the official POC efficacy testing, seven male monkeys were treated with single daily doses of triptonide (0.1 mg/kg BW) and three control male monkeys received vehicle. Since we already knew that significant effects on sperm production started at week 4, semen and blood samples were collected weekly starting from week 5. By the end of week 8, triptonide treatment was stopped in three of the seven monkeys, and the other four continued with triptonide treatment until week 126 (~2.4 years). For the monkeys undergoing long-term treatment, semen and blood samples were collected at 11, 14, 16, 18, 23, 48, 74, 100, and 126 weeks for semen and blood chemistry analyses.

To test fertility, two out of the three monkeys with treatment stopped at week 8 were individually housed with two fertility-proven adult female monkeys (one male and one female per cage) for up to 1 year. Both female monkeys became pregnant after 3–4 months and each delivered a full-term baby. Two out of the four male monkeys undergoing long-term treatment were also housed with two fertility-proven adult female monkeys (one male and one female per cage) between weeks 8 and 126. None of the two females became pregnant during the entire treatment. Two of the three controls were housed with two fertility-proven adult female monkeys (one male and one female per cage) between weeks 8 and 100. Both female monkeys became pregnant after 3–4 months, but one ended up with abortion for unknown reasons, but the other delivered a full-term baby.

For semen collection, a rectal probe electrical stimulation method was conducted as previously described[45] with minor modifications. Monkeys were anesthetized with ketamine hydrochloride (Lianyungang International Trade Co., Ltd., Lianyungang City, China) at a dose of 10 mg/kg BW through intramuscular injection. The monkeys were then held in a supine position on an operating table and penis was cleaned using absorbent cotton soaked with warmed saline. A lubricated probe was inserted into rectum for ~8–10 cm and positioned with both electrodes oriented in mid-ventral direction. A stimulator with nine-step voltage control (Lane Manufacturing Inc., USA) was used, and each electrical stimulation consisted of pulses of 4–6 s duration with 2–3 s rest in between. Stimulation started at step 1 for five repeats. If no painful response was observed, stimulation was continued with the voltage raised to next step. The stimulation was repeated and gradually intensified until ejaculation occurred. The ejaculates were collected into a tube and incubated in 37 °C water bath for 30 min before analyses. Semen samples were diluted using a sperm culture medium (K-SISM-20/50/10020, Cook Medical) to an appropriate concentration, and an aliquot of 10 μl was added to a pre-warmed (37 °C) Makler counting chamber and sperm counts, total and forward motility were analyzed manually by an experienced evaluator. For motility evaluation, at least 300 sperm were analyzed. For sperm morphology analyses, sperm smear was prepared. After air dry, the slides were fixed in 95% ethanol for 15 min at room temperature, followed by staining using SpermBlue fixative and SpermBlue stain[45].

To analyze testicular histology, testicular biopsy was performed under anesthesia (ketamine hydrochloride at 10 mg/kg BW i.m.) and sterile conditions. A small piece of testis (~3 mm × 3 mm × 3 mm) was surgically removed, and immediately fixed in Bouin's fixative. After dehydration, the samples were embedded into paraffin and sections of 4 μm were then stained using hematoxylin-eosin followed by microscopic evaluation.

Blood oxygen saturation levels were measured using a pulse oximeter (Yuwell, YX303, Yunyue Medical Equipment, Ltd, Jiangsu, China) by clamping the oximeter to the tail of the monkey. Blood assays included counts of the red blood cells, lymphocytes, mean corpuscular volume, and levels of prothrombin, hemoglobin, and glucose. The liver panel contained the following: albumin-globulin, albumin, alkaline phosphatase, alanine aminotransferase, aspartate aminotransferase, carbon dioxide, gamma-glutamyl transferase, and total protein. The kidney panel consisted of creatinine, urine albumin, and urea. All the assays were conducted in the Central Clinical Laboratory of the Hospital affiliated to the Institute of Family Panning of Guangdong Province (Guangzhou, China) using the reagents, equipment, and methodologies for human blood lab work. In brief, blood cell types and biochemistry were analyzed using a hematology analyzer (Sysmex XN-1000, Japan). FSH, LH, and T levels were measured using fluorescent immunoassays on an immunology analyzer (Roche Cobas E602, Switzerland). Liver and kidney panels were analyzed using a clinical chemistry analyzer (Olympus, AU400, Japan). Assays were conducted using reagents and protocols provided by the manufactures of the equipment.

**Determination of 50% of effective concentration.** $EC_{50}$ represents the dose at which a compound produces half of its maximal effect[46]. To determine the $EC_{50}$ of triptonide, a series of dose–response data were collected using adult male mice treated with various doses of triptonide in three independent experiments. Sperm motility was evaluated using CASA (Hamilton Throne). To determine male fertility, two to three fertility-proven females were added to the cages containing a single triptonide-treated or control male mouse at the end of the fourth weeks of treatment. Vaginal plugs were examined early every morning to verify mating, and pregnancy became visible by day 10 after mating. Later, the number of pups borne

was documented (Supplemental Table S3). $EC_{50}$ was calculated based on a basic rectilinear equation: $E = mx + E_0$, where $E$ represents the triptonide effect, $m$ stands for the slope $\left(\frac{E_0 - E_{0.1}}{0 - 0.1}\right)$, and $x$ for triptonide concentration ($C$). Since $E = \frac{E_0 - E_{0.1}}{0 - 0.1} C + E_0$, when $x = 0$, the $E$ equals $E_0$ ($E = E_0$). Because $E = \frac{E_0 - E_{0.1}}{0 - 0.1} C + E_0$, where $E$ 0.1 equals $E$, when $C = 0.1$. When $E = \frac{E_0}{2}$, the $C = C_{0.5}$. Since $E = \left(\frac{E_0 - E_{0.1}}{0 - 0.1}\right) C + E_0$; $\frac{E_0}{2} = \frac{E_0 - E_{0.1}}{0 - 0.1} C_{0.5} + E_0$, and $\frac{E_0}{2} = M C_{0.5} + E_0$. Thus, $EC_{0.5} = \frac{\frac{E_0}{2} - E_0}{M} = \frac{-\frac{E_0}{2}}{M} = -\frac{E_0}{2M}$. Specifically, we utilized $E_0$ (sperm motility or male fertility of control male mice) through its slope $\left(\frac{E_0 - E_{0.1}}{0 - 0.1}\right)$, and the equation ($C_{0.5} = -E_0/2m$), to acquire the $EC_{50}$.

**Western blot analyses**. Testes were lysed in Pierce IP lysis buffer (Thermo Scientific, no.87787, 25 mM Tris HCl, pH 7.4, 150 mM NaCl, 1% NP-40, 1 mM EDTA, and 5% glycerol) containing protease inhibitors (Roche, mini-complete, no EDTA) and homogenized. The lysates were vortexed and spun at $17,000 \times g$, 20 °C for 10 min, and the supernatants were collected. The protein concentrations were determined using Pierce BCA Protein Assay Kit (Thermo Scientific). The proteins from each sample were loaded onto one MiniProtean TGX 4–15% or 4–20% polyacrylamide gel (Bio-Rad) followed by electrophoresis at 200 V for 40 min. The gel was electroblotted onto nitrocellulose membrane at 100 V for 1 h in a cold room. After blocking with SuperBlock T20 (PBS) Blocking Buffer (Thermo Fisher Scientific) and the membrane was subjected to reactions with primary antibodies on a rocking platform in a cool room overnight. After washing, the membrane was incubated with goat anti-mouse IgG HRP or goat anti-rabbit IgG HRP for 1 h. The specific proteins were visualized using the Advanced Bright Enhanced Chemiluminescence kit. Imaging and qualification were performed using a ChemiDoc Imager Detector (Bio-Rad). The primary antibodies used included mouse monoclonal anti-γH2AX (phospS139, Abcam, Cat#: 2635), rabbit polyclonal anti-β-actin (Abcam, Cat#: 8227), rabbit polyclonal keratin 5 (K5; BioLegend, Cat#:905501), and mouse monoclonal anti-junction plakoglobin/gamma catenine (Life Technologies, Cat#:13-8500). Goat anti-rabbit (SouthernBiotech, Cat#: 4030-05) and goat anti-mouse IgG (H + L; SouthernBiotech, Cat#: 1036-05) HRP-conjugated secondary antibodies were purchased from SouthernBiotech. Original uncropped western blot scans or films can be found in the Source data file.

**Drug affinity responsive target stability assay**. Four testes were removed from adult male C57/BL/6J mice. Each testis was dissected into several pieces and put into a 1.5 ml tube containing 450 μl M-PER solution (Thermo Fisher Scientific) with protease inhibitors (Roche, complete mini, EDTA-free) and phosphatase inhibitors (Abcam, Phosphatase inhibitor cocktail I). The mixture was homogenized with 20 strokes of a plastic pestle and then homogenized with a Benchmark D1000 homogenizer at a setting of two for 15 s. All four testis samples sat on ice for a half hour and were then centrifuged at $17,200 \times g$ for 10 min. The four supernatants were pooled in a 15 ml centrifuge tube. To the tube was added 200 μl 10× TNC (500 mM Tris HCl, 500 mM NaCl, 100 mM CaCl₂, pH 8.0). The homogenate was assayed for protein concentration with a BCA assay and was diluted to 5 mg/ml. Homogenate (297 μl) was aliquoted into two tubes. To the tubes were added either 3 μl 16 mg/ml triptonide in DMSO or DMSO alone. The tubes were mixed well and left at room temperature for 1 h. Meanwhile, a solution of 10 mg/ml Pronase was diluted as follows: 1:100, 1:300, 1:1000, 1:3000, and 1:10,000. When Thermolysin was used in place of Pronase, a 10 mg/ml solution of Thermolysin was diluted to 2, 0.4, 0.08, and 0.016 mg/ml. The triptonide-treated or control homogenate (50 μl) was incubated with each of trypsin dilution for 15 min. The reactions were then stopped by the addition of 5 μl 0.5 M EDTA. Each sample was then run on Bio-Rad mini-Protein TGX 4-20% polyacrylamide gels at 200 V. Gels were stained with Bio-Safe Coomassie (Bio-Rad).

An in vivo variation of the DARTS experiment was carried out using testis homogenates from mice treated with 0.8 mg/kg BW triptonide daily (p.o.) for 4 weeks. Protein was precipitated from these extracts with four volumes of ice-cold acetone. After overnight incubation at −20 °C, the samples were centrifuged at $16,000 \times g$, 4 °C, for 10 min. The pellet was washed three times with ice-cold acetone/water (4:1). The protein pellet was dried, dissolved, and digested for various lengths of time using the Flash Digest system from Perfinity Biosciences. The digest reaction was stopped after 0, 5, 15, and 75 min of incubation by removal of aliquots and addition into Laemmli buffer with β-mercaptoethanol. The aliquots were run on polyacrylamide gels followed by visualization of protein using staining. Bands of interest were cut out and submitted for MS analyses. Original gel images can be found in the source data file.

**GlycoLink beads-based affinity purification**. GlycoLink Micro Immobilization Kit was purchased from Thermo Fisher Scientific. GlycoLink beads contain a linker with a terminal hydrazide group which reacts with aldehydes and ketones. The coupling of triptonide to the beads was attempted in two ways. The GlycoLink kit comes with an acidic coupling buffer for coupling to aldehydes and ketones. An additional basic buffer was purchased which might couple via the epoxide groups of triptonide. Coupling was done by incubating beads with a saturated solution of triptonide in DMSO (40 mg/ml), or with DMSO alone for controls, in acidic and basic buffer. Triptonide was covalently attached to the beads with aniline according

to the manufacturer's instructions. Testes from one sexually mature male mouse were homogenized in M-PER buffer (Thermo Fisher Scientific). The proteins in the homogenate were precipitated with four volumes of ice-cold acetone. After sitting for 3.5 h at −20 °C, the protein pellet was centrifuged at $16,000 \times g$, 4 °C, for 10 min. The pellet was washed three times with ice-cold 4:1 acetone/water. The final pellet was dried and dissolved in 2.5 ml Dulbecco's PBS. An aliquot (300 μl) was applied to each of the four types of beads and reacted with the beads as per instructions. After washing of the beads, proteins on the beads were eluted sequentially with the following solutions: a saturated solution of triptonide in DMSO in PBS, 0.1% formic acid/30% acetonitrile, and 0.2 M glycine, pH 2.6. The eluates were dried in a vacuum concentrator to ~60–70 μl. The eluates were then desalted with Zeba Spin Desalting Kit (Thermo Fisher Scientific, 7 K MWCO) and buffer exchanged into 0.1% formic acid/30% acetonitrile. The eluates were again dried to a volume of 35–40 μl. The eluates were then applied to a Bio-Rad mini-Protean TGX 4-20% polyacrylamide gels and run at 200 V. Gels were stained with Sypro Ruby. Bands of interest were cut out and submitted for MS analyses. Original gel images can be found in the Source data file.

**Immunoprecipitation-based identification of proteins interreacting with the N-terminus of SPEM1**. A peptide composed of the N-terminal 28 amino acids of murine SPEM1 was synthesized by BioLegend (San Diego, CA). A biotin was attached to the C-terminal amino acid of the peptide. Streptavidin magnetic beads were obtained from GenScript. Beads were incubated with either 0.3 mg of peptide in PBS or with PBS alone. The solutions were incubated with the beads for 1 h and then washed with PBS. A homogenate of testis from a sexually mature male mouse was prepared in PBS with protease inhibitors (Roche, complete mini, EDTA-free). The two types of beads were incubated with 120 μl of the testis homogenate (0.34 mg protein) for 1 h. After washing of the beads, proteins on the beads were eluted sequentially with a saturated solution of triptonide in DMSO in PBS, 0.1% formic acid/30% acetonitrile, and 0.2 M glycine, pH 2.6. The eluates were dried in a vacuum concentrator to ~60–70 μl. The eluates were then desalted with Zeba Spin Desalting Kit (Thermo Fisher Scientific, 7 K MWCO) and buffer exchanged into 0.1% formic acid/30% acetonitrile. The eluates were again dried to a volume of 35–40 μl. Half of each eluate was diluted with Laemmli buffer with β-mercaptoethanol and applied to a Bio-Rad mini-Protean TGX 4–20% polyacrylamide gels and run at 200 V. Gels were stained with Sypro Ruby. The remaining halves of the eluates were submitted for MS analysis.

**Mass spectrophotometric identification of proteins**. Proteins in the excised electrophoresis bands were reduced and alkylated using 10 mM dithiothreitol and 100 mM iodoacetamide. Proteins in solutions were denatured with acetonitrile, then reduced and alkylated with 6.7 mM DTT and 13.8 mM iodoacetamide. The proteins were then incubated with sequencing grade modified porcine trypsin (Promega, Fitchburg, WI) in 25 mM ammonium bicarbonate overnight at 37 °C.

*Liquid chromatography*. Peptide mixtures were separated using an UltiMate 3000 RSL Cnano system (Thermo Scientific, San Jose, CA) on a self-packed UChrom C18 column (100 μm × 35 cm). Elution was performed using a 90 min gradient of solvent B from 2–27% (solvent A 0.1% formic acid, and solvent B acetonitrile, 0.1% formic acid) at 50 °C using a digital Pico View nanospray source (New Objectives, Woburn, MA) that was modified with a custom-built column heater and an ABIRD background suppressor (ESI Source Solutions, Woburn, MA). Briefly, the self-packed column tapered tip was pulled with a laser micropipette puller P-2000 (Sutter Instrument Co, Novato, CA) to an approximate id of 10 μm. The column was packed with 1–2 cm of 5 μm Magic C18 followed by 35 cm of 1.8 μm UChrom C18 (120 A) at 9000 p.s.i., using a nano LC column packing kit (nanoLCMS, Golg River, CA).

*Mass spectrometry*. Mass spectral analysis was performed using an Orbitrap Fusion mass spectrometer (Thermo Scientific. San Jose, CA). Proteomic analysis was performed using a "Universal" data-dependent method (Eliuk et al. 2014 Thermo Application note). The MS1 precursor selection range was from 400 to 1500 $m/z$ at a resolution of 120 K and intensity threshold of $4.0 \times 10^5$. Quadrupole isolation at 0.7T h for MS² analysis using CID fragmentation in the linear ion trap with a collision energy of 35%. The automatic gain control was set to $1.0 \times 10^7$ with a maximum injection time of 250 ms.The instrument was set in a top speed data-dependent mode with a most intense precursor priority. Dynamic exclusion was set to an exclusion duration of 60 s with a 10 ppm tolerance.

*Database searching*. Tandem mass spectra were extracted and charge state deconvoluted by Proteome Discover version 2.1. All MS/MS samples were analyzed using Sequest (Thermo Fisher Scientific, San Jose, CA, USA; version 2.0.0.802). Sequest was set up to search a custom C-term fasta, assuming the digestion enzyme trypsin and max number of missed cleavages get to 2. Sequest was searched with a fragment ion mass tolerance of 0.60 Da and a parent ion tolerance 10.0 PPM. Variable modifications included carbamidomethyl of cysteine, oxidation of methionine, deamidation, and acetylation of the N-terminus.

*Criteria for protein identification.* Scaffold (version Scaffold 4.8.2, Proteome Software Inc., Portland, OR) was used to validate MS/MS-based peptide and protein identification. Peptide identifications were accepted if they could be established at >80% probability by the Peptide Prophet algorithm[47] with Scaffold delta-mass correction. Protein identifications were accepted if they could be established at greater 99.0% probability and contained at least two identified peptides. Protein probabilities were assigned by the Peptide Prophet algorithm[48]. Proteins that contained similar peptides and could not be differentiated based on MS/MS analysis alone were grouped to satisfy the principles parsimony.

**Statistics and reproducibility**. Graph Pad Prism 7 software (La Jolla, CA, USA) was used for statistical analyses. Data are presented as mean ± SEM unless stated otherwise. Differences in measurements were compared by Kolmogorov–Smirnov *t* test or one-way analyses of variance (ANOVA) for two groups, and $p < 0.05$ was considered statistically significant. Two-way ANOVA with Bonferroni multiple comparison test were used to compare differences between groups and timepoints, and adjusted $p < 0.05$ was considered to be statistically significant.

Gross morphology and histology of both the testes and cauda epidydimal sperm were routinely examined in both pilot and POC efficacy testing experiments using C57Bl/6J (approximately ten independent experiments expanding 7 years) and CD-1 (two independent experiments in two years) male mice, and results similar to the representative images shown in Figs. 1a, b and Supplementary Fig. 2 were obtained consistently. Both TEM and SEM were conducted once using three mice from three separate experiments, and the data similar to the representative images shown in Fig. 1c were obtained consistently. In the POC efficacy testing on monkeys, ejaculated sperm were collected from all monkeys once every 1–3 weeks, and testicular biopsy was conducted on six treated and three control monkeys at different timepoints. Both sperm morphology and testicular histology varied to some degree, but were generally consistent to the representative images shown in Fig. 2a, e, as well as Supplementary Fig. 4d–i. Pathology examinations at both gross and histological levels were conducted on six treated and three control male C57Bl/6J mice from three independent experiments, and no major pathology was detected and the histology of major vital organs of these mice was similar to the micrographs shown in Supplementary Fig. 3. DARTs were performed twice with similar results, and Fig. 5b is one the two gels obtained and subjected to MS-based protein identification. GlycoLink beads-based affinity purification was repeated four times using samples from four male C57Bl/6J male mice from two independent experiments, and results similar to Fig. 5d–f were obtained. Immunofluorescent and western blot analyses of gamma-H2Ax were carried out three times, using three control and six treated male mice from two independent experiments, and results similar to those shown in Supplementary Fig. 14 were obtained. Efficacy testing using chemically synthesized triptonide was performed twice, using four mice per group per timepoint, and both testicular histology and sperm morphology were similar to those shown in Supplementary Fig. 16.

**Reporting summary**. Further information on research design is available in the Nature Research Reporting Summary linked to this article.

## Data availability

All data are available within the Article and Supplementary Files, or available from the corresponding author on reasonable request. Source data are provided with this paper.

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

## Acknowledgements

We are grateful to Dr. John R. McCarrey for critical reading of the manuscript. Drs. Thomas Bell and Ali Rameez are acknowledged for their help with triptonide purity analyses. This study was supported by grants from the XtraThordinary, LLC. and the Male Contraceptive Initiative (to W.Y.). The POC efficacy testing on cynomolgus monkeys was supported by grants from the National Natural Science Foundation of China (No. 81801523 to Y.Z.), the Natural Science Foundation of Guangdong Province (2015A030313884 to Y.T.; 2018A030313528 and 2019A1515011984 to W.Q.), Guangdong Province Medical Research Funding (No. A2018297 to W.Q., A2018075 to J. W., and A2018235 to X.L.), the Science and Technology Planning Foundation of Guangzhou City (201607010137 to S.D. and 201904010017 to Y.Z.), and the Family Planning Research Institute of Guangdong Province (S2014001 to Y.T., S2018004 to J.W., S2018012 to H.l., and S2018013 W.Q.). The University of Virginia Center for Research in Reproduction Ligand Assay and Analysis Core is supported by a National Centers for Translational Research In Reproduction and Infertility (NCTRI) grant from the NICHD (P50-HD28934).

## Author contributions

W.Y. conceived and designed the overall study. W.Y. and H.Z. supervised all of the mouse studies conducted at the University of Nevada, Reno School of Medicine. W.Y., Y.T., and W.Q. supervised all of the testing on cynomolgus monkeys. Z.C., H.Z., K.S., Y.W., H.M., Z.W., H.P., and S.Y. conducted all of the experiments on mice, whereas W.Q., X.L., J.W., Y.W., S.Z., Y.J., H.N., Y.T., and Y.Z. performed all of the experiments on cynomolgus monkeys. M.J.M.H provided advices on the POC efficacy testing. L.H. performed all of the statistical analyses on data from monkeys. W.Y. wrote the manuscript. All authors reviewed and agreed on the contents of the manuscript.

## Competing interests

The authors declare no competing interests.
