## [Peer Review File · Nature Communications]

Reviewers' Comments:

Reviewer #1:

Remarks to the Author:

There are still several remaining outstanding issues with the questions and responses below: :

1. Intratesticular testosterone is not simply assessed by measuring a cytosol or homogenate of the testis. It is a laborious task where the steroids are extracted, precipitated with inclusion of a tracer, resuspended and then analyzed or by assay after HPLC (Chung JY, Chen H, Midzak A, Burnett AL, Papadopoulos V, Zirkin BR. Drug ligand-induced activation of translocator protein (TSPO) stimulates steroid production by aged brown Norway rat Leydig cells. *Endocrinology*. 2013;154(6):2156-2165. doi:10.1210/en.2012-2226 and Handelsman DJ, Jimenez M, Singh GK, Spaliviero J, Desai R, Walters KA. Measurement of testosterone by immunoassays and mass spectrometry in mouse serum, testicular, and ovarian extracts. *Endocrinology*. 2015;156(1):400-405. doi:10.1210/en.2014-1664 for examples). Without extraction using commercially available tests the majority of results are not accurate.

Reply: We had our samples analyzed in the Center for Research in Reproduction, University of Virginia School of Medicine, Charlottesville, Virginia. This center has been supported by the NICHD for many years and serves as a centralized resource for hormone (LH FSH, Testosterone, Progesterone and estradiol) measurement in both mice and humans. The methods that they used are similar to what this reviewer suggested.

Comment: as an expert in testosterone analysis and the challenges of qualifying testosterone concentrations --and a past user of the U. Virginia Sch. Med. Charlottesville, VA, the NICHD supported testing facility, I have first hand knowledge that this facility does not (unless they added a new method for testosterone measurement from tissues using either mass spec or an extraction method with recovery correction within the last 6 months) perform methods appropriate for testosterone analysis on tissue extracts. Here is another paper on the problems of testosterone measurement in testis and ovarian extracts that stresses that the validity of direct (non-extraction) testosterone immunoassays developed for use with human serum samples results in measurement of testosterone in almost all samples but a minority provided an accurate result in serum- tissue extracts have even more challenges . The Chung paper stresses the need for alternative methods which they employed to ensure the results were correct. Please determine the method employed and this should be stated in the methods section.

But there is significant concern that the values shown in the Fig. 1 are too low even for C46Bl/6 male mice- given the Y axis off Fig. 1 it is difficult to discern how low this value is as it is plotted for control mice to be less than 1 ng/mL for many with just 2 of the 11 appearing to be measurable- the limits of detection of the asset should be specified and the axis adjusted to allow accurate representation of the data and still account for the outlier (dominant male). Others report C57Bl/6 J males have about 1 ng/mL or 100 ng/dL (Sex and strain differences in adult mouse cardiac repolarization: importance of androgens Judith Brouillette, Katy Rivard, Eric Lizotte, Céline Fiset Author Notes

Cardiovascular Research, Volume 65, Issue 1, January 2005, Pages 148–157,

<https://doi.org/10.1016/j.cardiores.2004.09.012>)

which is approximately what our laboratory found (~140ng/dL T) in circulation .

Measurement of Testosterone by Immunoassays and Mass Spectrometry in Mouse Serum, Testicular, and Ovarian Extracts David J. Handelsman, Mark Jimenez, Gurmeet K. S. Singh, Jenny Spaliviero, Reena Desai, Kirsty A. Walters
Endocrinology, Volume 156, Issue 1, 1 January 2015, Pages 400–405,
<https://doi.org/10.1210/en.2014-1664>

Testosterone is usually expressed as ng/dL. As presented the values are too low- if the males were not housed separately (for the mice), it is common for there to be dominant and submissive males with extremely high levels for the dominant male and very low levels for the submissive males. As plotted, it is impossible to see what is happening with this data (Figure 1) and the housing should have been individual with no female mice in the room to avoid these issues.

Reply: Agreed. We were fully aware of this possibility and thus, always individually housed the male mice (one male mouse per cage). In fact, our animal facility did not allow us to pool them to avoid fighting and its related stress and injuries. The units were provided by the Core Lab at University of Virginia School of Medicine.

Comment: The serum T shown looks to be about 1 ng/mL but normal ranges in mice (which vary somewhat by strain- see figure 1 in Sex and strain differences in adult mouse cardiac repolarization: importance of androgens Judith Brouillette, Katy Rivard, Eric Lizotte, Céline Fiset Author Notes Cardiovascular Research, Volume 65, Issue 1, January 2005, Pages 148–157, <https://doi.org/10.1016/j.cardiores.2004.09.012>)

Reviewer #2:

Remarks to the Author:

My concerns have been addressed.

Reviewer #3:

Remarks to the Author:

The authors have addressed the criticisms of this reviewer and the manuscript is improved greatly. The submission addresses an important question and provides an excellent roadmap for this research topic.

Point-by-point response to the reviewers' comments

Reviewer #1 (Remarks to the Author):

There are still several remaining outstanding issues with the questions and responses below: :

1. As an expert in testosterone analysis and the challenges of qualifying testosterone concentrations -- and a past user of the U. Virginia Sch. Med. Charlottesville, VA, the NICHD supported testing facility, I have first-hand knowledge that this facility does not (unless they added a new method for testosterone measurement from tissues using either mass spec or an extraction method with recovery correction within the last 6 months) perform methods appropriate for testosterone analysis on tissue extracts. Here is another paper on the problems of testosterone measurement in testis and ovarian extracts that stresses that the validity of direct (non-extraction) testosterone immunoassays developed for use with human serum samples results in measurement of testosterone in almost all samples, but a minority provided an accurate result in serum- tissue extracts have even more challenges . The Chung paper stresses the need for alternative methods which they employed to ensure the results were correct. Please determine the method employed and this should be stated in the methods section.

Reply: I totally agree with this reviewer that accurate measurement of T levels is indeed challenging. When I was a graduate student working with Prof. Ilpo Huhtaniemi in Finland ~23 years ago, he always emphasized the difficulties in T measurement. All of our samples were prepared using ether extraction and he only trusted 1 or 2 places in Europe for the radio-isotope-based immunoassays. The Core Lab at Univ. Virginia used the Testosterone ELISA Kit from IBL (Cat#IB79106). We did use ether extraction method to prepare testicular samples initially, but were told NOT to use ether extraction because the kit is not compatible with this method. Instead, we were given their protocol to prepare the testicular homogenates for the assays.

While we appreciate the concerns over the accuracy of the assays conducted by the Core Lab, we do believe the assay results appear to truthfully reflect the relative testosterone levels for the following reasons:

- 1) It is well-known that intra-testicular T levels are much greater than the circulating/serum T levels. We did see this in the ELISA assays. We had to dilute the testicular homogenate samples by a minimum of 10-100X in order to keep the readings within the reportable range (10-1,600ng/dL). Here is an example of the original data report that we received from the NIH-supported Core Lab at Univ. Virginia:

Testosterone Mouse & Rat IBL ELISA		File: TESMR_021519A	
Investigator: Wei Yan/Zheng		Reportable Range= 10.0 - 1600.0 ng/dL	
Samples received: 12/19/18			
Assay Date: 2/15/19			
Kit Lot 29K118-2			
Rodent QC1-Lot 112117; QC2-Lot 022516; QC3 Lot 061218			
Standard Lot 29S088			
Tech NVG	Replicates	MEAN	% CV
SAMPLE ID	(ng/dL)	(ng/dL)	
Rodent QC1 (32.0-60.0 ng/dL)		43.6	0.8
Rodent QC 2 (78.0 - 146.0 ng/dL)		105.6	0.3
Rodent QC 3 (510.0 - 950.0 ng/dL)		732.9	3.7
Wei Yan/Zheng (12/19/18)-1		144.5	
2		76.8	
3		39.1	
4		1376.9	
5		57.6	

Point-by-point response to the reviewers' comments

- 2) We did observe that the dominant males tended to have higher T levels when samples from a group of 4 mice from the same cage (maximal number of mice allowed per cage in our facility) were measured. We would like to clarify that in our study, the treated male mice were individually caged (i.e., one male mouse per cage) throughout the experiments because we needed to add 2-3 females into the cages after 4 weeks of treatment to test fertility of the male mice.
- 3) Although the final readings/numbers may differ from those in the classic methods, both control and treated samples were analyzed using exactly the same procedure and reagents. Therefore, the data should be theoretically comparable at least between control and treated samples in our study.
- 4) We have monitored the sexual behavior of those treated male mice, e.g., mounting, and found no differences between control and treated male mice.
- 5) The strongest evidence supporting no changes in T levels in treated males came from the fact that the mating efficiency was similar between control and treated males because a similar number of plugged females were obtained in our large-scale fertility tests for determining EC₅₀ (Figure 4 and Supplemental table 3).
- 6) No major differences in T levels were observed in treated male monkeys either (Supplemental Figure 6).

Following suggestions of this review, we added more details regarding the methods used for mouse hormonal assays (Page 21, lines 16-26; page 22 lines 1-3).

2. But there is significant concern that the values shown in the Fig. 1 are too low even for C57Bl/6 male mice- given the Y axis of Fig. 1 it is difficult to discern how low this value is as it is plotted for control mice to be less than 1 ng/mL for many with just 2 of the 11 appearing to be measurable- the limits of detection of the assay should be specified and the axis adjusted to allow accurate representation of the data and still account for the outlier (dominant male). Others report C57BL/6 J males have about 1 ng/mL or 100 ng/dL (Sex and strain differences in adult mouse cardiac repolarization: importance of androgens Judith Brouillette, Katy Rivard, Eric Lizotte, Céline Fiset Author Notes Cardiovascular Research, Volume 65, Issue 1, January 2005, Pages 148–157, <https://doi.org/10.1016/j.cardiores.2004.09.012>), which is approximately what our laboratory found (~140ng/dL T) in circulation. Measurement of Testosterone by Immunoassays and Mass Spectrometry in Mouse Serum, Testicular, and Ovarian Extracts David J. Handelsman, Mark Jimenez, Gurmeet K. S. Singh, Jenny Spaliviero, Reena Desai, Kirsty A. Walters Endocrinology, Volume 156, Issue 1, 1 January 2015, Pages 400–405, <https://doi.org/10.1210/en.2014-1664>

The serum T shown looks to be about 1 ng/mL but normal ranges in mice (which vary somewhat by strain- see figure 1 in Sex and strain differences in adult mouse cardiac repolarization: importance of androgens Judith Brouillette, Katy Rivard, Eric Lizotte, Céline Fiset Author Notes Cardiovascular Research, Volume 65, Issue 1, January 2005, Pages 148–157, <https://doi.org/10.1016/j.cardiores.2004.09.012>)

Reply: We re-examined all of the original data and found that three values were mistakenly used directly without being converted to ng/ml. Since the unit used in the original report was ng/dL, these values were 100x greater than those in ng/mL, causing the y-axis to be super long; consequently, those converted

Point-by-point response to the reviewers' comments

values were compacted together, leading to the difficulty in reading the actual values in previous Fig 1j. I sincerely apologize for this error. Thanks to this reviewer for catching this problem! We now have re-organized the data and adopted ng/dL as the unit, as suggested by this reviewer, and made new Fig. 1j. We also thoroughly re-examined the data on testicular T, FSH and LH and replotted the data, which are shown in Fig. 1i, k and l. We also provided all raw data for all figures in a file titled "Source Data".

Reviewer #2 (Remarks to the Author):

My concerns have been addressed.

Reviewer #3 (Remarks to the Author):

The authors have addressed the criticisms of this reviewer and the manuscript is improved greatly. The submission addresses an important question and provides an excellent roadmap for this research topic.